# EasyParallel: A GUI platform for parallelization of STRUCTURE and NEWHYBRIDS analyses

**Honggang Zhao[1]\*, Benjamin Beck[2], Adam Fuller[3], Eric Peatman[1]**

**1** School of Fisheries, Aquaculture, and Aquatic Sciences, Auburn University, Auburn, AL, United States of America, **2** Agricultural Research Service, Aquatic Animal Health Research Unit, United States Department of Agriculture, Auburn, AL, United States of America, **3** Agricultural Research Service, Stuttgart National Aquaculture Research Center, United States Department of Agriculture, Stuttgart, AR, United States of America

\* hzz0024@auburn.edu

**Data Availability Statement:** The software, source code, user manual, and example data sets are available online from https://github.com/hzz0024/EasyParallel

## Abstract

The software programs STRUCTURE and NEWHYBRIDS are widely used population genetic programs useful in addressing questions related to genetic structure, admixture, and hybridization. These programs usually require a large number of independent runs with many iterations to provide robust data for downstream analyses, thus significantly increasing computation time. Programs such as Structure_threader and parallelnewhybrid were previously developed to address this problem by processing tasks in parallel on a multi-threaded processor; however some programming knowledge (e.g., R, Bash) is required to run these programs. We developed EasyParallel as a community resource to facilitate practical and routine population structure and hybridization analyses. The multi-threaded parallelization of EasyParallel allows processing of large genetic datasets in a very efficient way, with its point-and-click GUI providing ready access to users who have little experience in script programming. Performance evaluation of EasyParallel using simulated datasets showed similar speed-up and parallel execution time when compared to Structure_threader and Parallelnewhybrid. EasyParallel is written in Python 3 and freely available on the GitHub site https://github.com/hzz0024/EasyParallel.

## 1. Introduction

Recent advances in next-generation sequencing (NGS) technologies and the decreased cost of NGS have led to a rapid accumulation of genetic data for both model and non-model organisms [1]. To accommodate this data explosion, new tools and computation platforms were developed to perform parallelized data analyses [2,3]. However, most of these programs were compiled and executed in command-line based environments (e.g., Linux, R), which could make them less accessible and appealing to users who have little programming background. Moreover, some programs require independent runs with many iterations to provide robust data for downstream analysis, making it time-consuming when the dataset includes a large number of individuals and genetic markers.

**Funding:** The authors received no specific funding for this work.

**Competing interests:** The authors have declared that no competing interests exist.

One such example is STRUCTURE [4]. This Bayesian-based clustering approach utilizes individual genotypes and population allele frequencies to cluster individuals, with the assumptions of Hardy–Weinberg and linkage equilibrium of marker loci within populations [4]. Since its publication, STRUCTURE has been widely applied to address questions related to population structure, species or individual assignment, hybridization and introgression [5–10]. Because STRUCTURE requires to minimize the effect of the starting configuration, many iterations are needed during the burnin process [6]. More importantly, STRUCTURE is usually run with many iterations for different genetic cluster values ($K$) to determine the optimal number of populations [11], thus significantly increasing computational times.

Another program requiring a large number of independent runs is NEWHYBRIDS [12]. Using Bayesian model-based clustering and MCMC simulation, NEWHYBRIDS computes the posterior probability of each individual that falls into distinct hybrid classes [12]. Although both programs were designed with graphical interfaces and cross-platform compatibility (Linux, Windows, and MacOS), the native GUIs do not facilitate multiple independent analyses. Additionally, parameters and input files must be copied and edited manually between runs, which introduces the potential for human errors [13]. To increase the efficiency and speed of running these programs, strategies such as parallel processing and script programming on multiple cores/threads have been previously proposed for STRUCTURE or NEWHYBRIDS analyses [13–16]. Although these strategies are invariably more convenient and efficient, some knowledge of programming languages is still needed.

The program EasyParallel presented in this article is provided as a free cross-platform tool that utilizes a multi-thread parallel algorithm for processing multiple iterations of STRUCTURE and NEWHYBRIDS analyses. EasyParallel employs a user-friendly graphical user interface (GUI) and multi-core parallelization for multiple independent runs of a dataset.

## 2. Materials and methods

### 2.1 Overview

EasyParallel is freely available at https://github.com/hzz0024/EasyParallel with installation instructions and a brief demo provided in the *Documentation* site. EasyParallel requires the command-line version of STRUCTURE and NEWHYBRIDS programs. Thus, a user must download the correct version of the target program and load the main directory (with executable files) to EasyParallel. Python is used for directory creation, data processing, parallel runs, and file writing operations. At present, EasyParallel can perform parallel replication runs for STRUCTURE and NEWHYBRIDS across MacOS and Windows operating systems, with all source code packaged for the direct run without installation. However, the open-source design of EasyParallel can be extended to other compatible software that requires multiple iterations or simulations for data analysis.

### 2.2 Parallel scheme

In order to achieve parallelism, one intuitive approach is to copy the entire folder $n$ times ($n$ is the number of the run) and run each copy in parallel. On the contrary, we use a "single executable multiple working directories" scheme–i.e., each subprocess executes the same executable file, but in different working directories. The "multiple working directories" design is implemented with the subprocess management (https://docs.python.org/2/library/subprocess.html) of Python Standard Library, a module which is able to set the child working directory before it is executed. The benefit of our design is two-fold: 1) we execute the software $n$ times in parallel without the necessity to make $n$ copies of the executable file. All the child processes share the same executable file, and produce the outputs in an independent directory; 2) EasyParallel

platform is not confined by output constraints (e.g., NEWHYBRIDS does not allow specification of an output directory and generates outputs into the working directory instead). In our design, such constraints are addressed by designated working directories.

### 2.3 User-friendly GUI

For the EasyParallel graphical user interface (GUI), we provide a progress bar and a window to show the status of parallelization (Fig 1). Because both STRUCTURE and NEWHYBRIDS require specific parameters for data running, the software interface for each module was designed to support parameter modification (e.g., number of repeats and threads used for parallel execution). In addition, the user could specify the location of additional datasets and parameter input files in an intuitive and convenient manner (e.g., drag *mainparams* and *extraparams* files directly to the EasyParallel GUI for STRUCTURE analysis). If not supplied by the user, the default settings of parameter files archived from the target program will be used.

### 2.4 Execution time analyses

We used two datasets available in Pina-Martins et al., [15] and Wringe [16] to evaluate the execution time and speed gain of EasyParallel in STRUCTURE and NEWHYBRIDS analyses, respectively. We used the GUI version for execution time analyses. Four laptops with various core architectures (2, 4, and 6 physical cores) and different operating systems (Windows and MacOS) were used for performance comparison: Lenovo Y510, Windows 10, 2.4 GHz Intel Core i7- 4700MQ with 8 GB RAM, 4 physical cores with 8 logical threads (i7 4700MQ); Lenovo Y700, Windows 10, 2.6 GHz Intel Core i7-6700HQ with 8 GB RAM, 4 physical cores

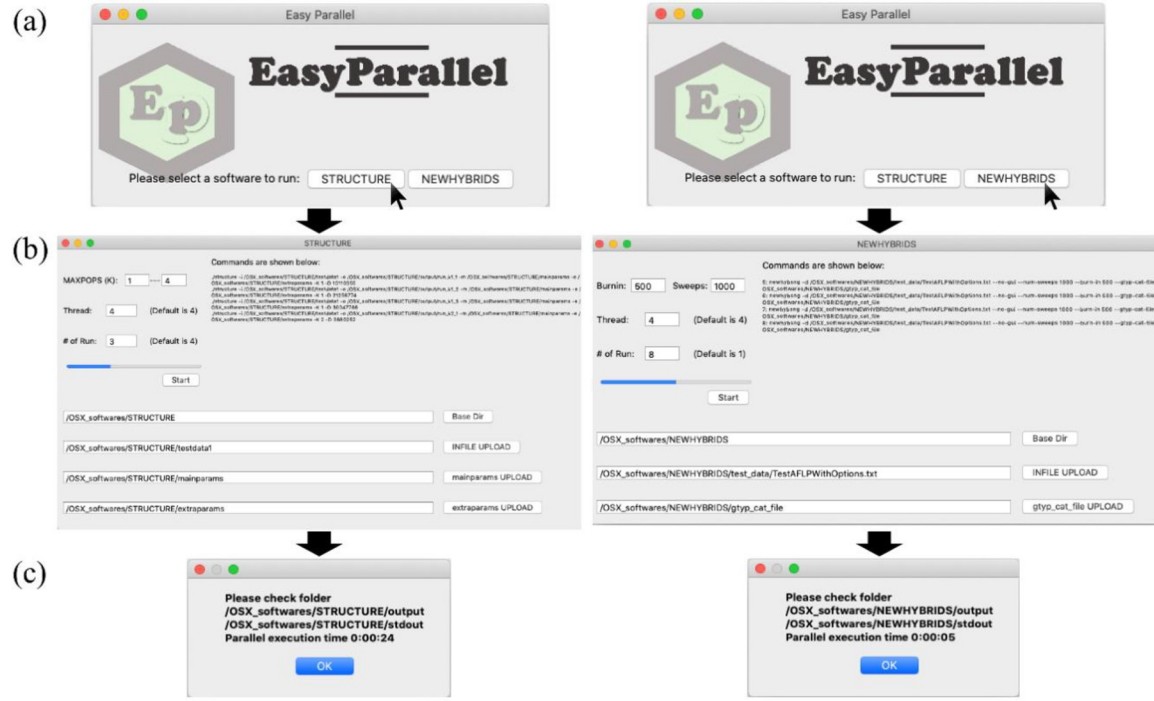

**Fig 1. A screenshot of EasyParallel running the STRUCTURE and NEWHYBRIDS analyses in parallel.** (a) EasyParallel main window allows user to choose the STRUCTURE or NEWHYBRIDS module for data analysis. (b) The module panel assisting the user in adding major parameters (e.g. the number of thread or runs) and the input/parameter files. A progress bar at left shows the status of parallelization. A command window at top right shows the commands used for data running. (c) Message window shows the folder storing the outputs and the time to complete the analysis.

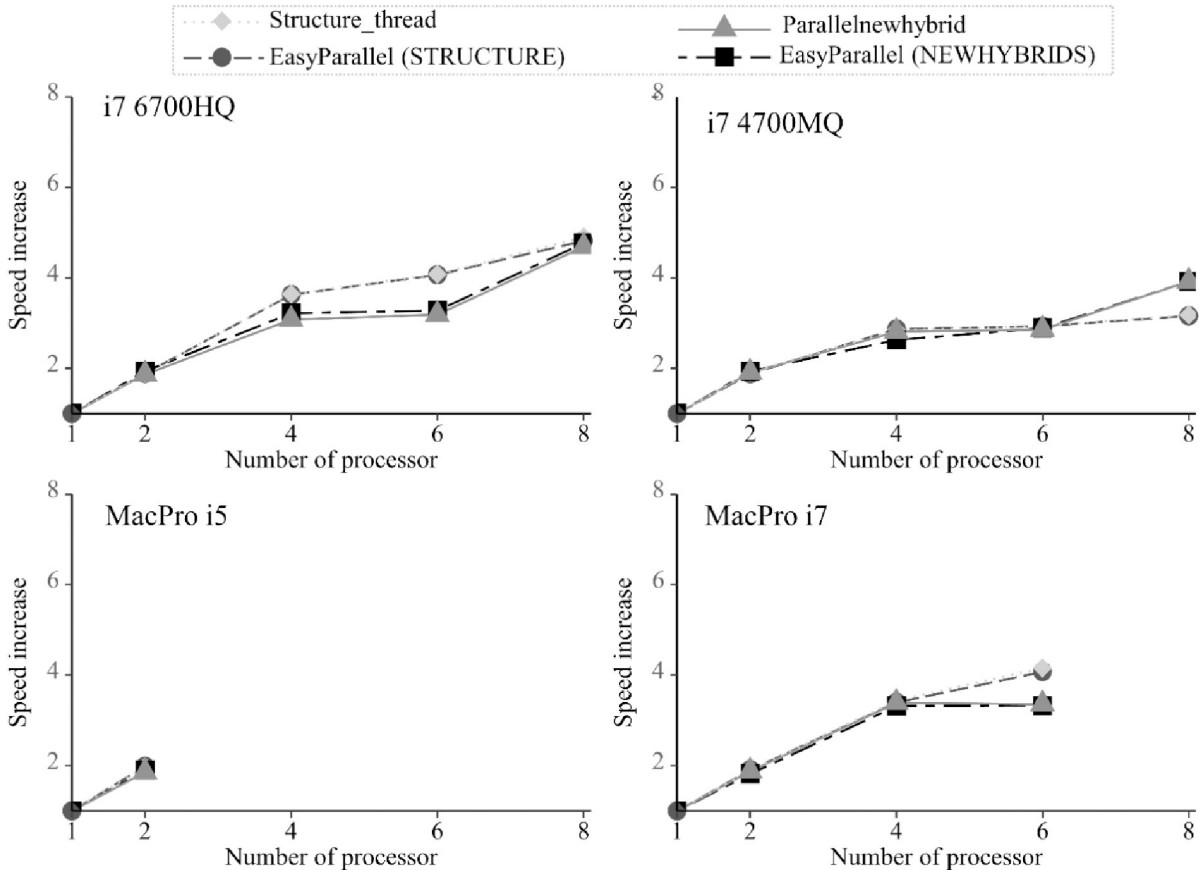

**Fig 2. Speed gain obtained by parallelization in EasyParallel and its comparison with Structure_threader and Parallelnewhybrid.** The speed increase was calculated by dividing the execution time on a single thread (sequential run) by the execution time obtained from different number of threads. i7 4700MQ – Lenovo Y510, Windows 10, 2.4 GHz Intel Core i7- 4700MQ with 8 GB RAM and 4 physical cores (8 logical threads); i7 6700HQ – Lenovo Y700, Windows 10, 2.6 GHz Intel Core i7-6700HQ with 8 GB RAM and 4 physical cores (8 logical threads); MacPro i5 – MacBook Pro, OS 10.14, 2.7 GHz Intel Core i5 with 16 GB RAM and 2 physical cores; MacPro i7 – MacBook Pro, OS 10.14, 2.6 GHz Intel Core i7 with 16 GB RAM and 6 physical cores.

with 8 logical threads (i7 6700HQ); MacBook Pro, OS 10.14, 2.7 GHz Intel Core i5 with 16 GB RAM, 2 physical cores (MacPro i5); MacBook Pro, OS 10.14, 2.6 GHz Intel Core i7 with 16 GB RAM, 6 physical cores (MacPro i7). The test file used for STRUCTURE analysis consisted of 100 individuals and 80 single nucleotide polymorphism (SNP) loci (total 8,000 genotypes with no missing data). This dataset was initially crafted based on data from the 1,000 Genomes Project (The 1,000 Genomes Project Consortium, 2015) and is available in the program's repository. STRUCTURE was run using the admixture model with correlated allele frequencies and $5 \times 10^4$ burn-in period followed by $1 \times 10^6$ Markov Chain Monte Carlo (MCMC) repeats. These settings were applied for values of $K$ ranging from 1 to 4, with four independent runs for each K (a total of 16 STRUCTURE runs). For NEWHYBRIDS, eight independent analyses were run on a simulated data set with 100 loci and 200 individuals for each of the six genotype frequency classes (pure1, pure2, F1, F2, BC1, and BC2), with an initial burn-in of 500 replicates and 1,000 MCMC sweeps afterward (following the same settings as Wringe et al, [16]). To assess the execution time obtained by parallelization in EasyParallel, we computed the "speed up" values using the equation of $S_p = T_1/T_p$, where $S_p$ is the speed-up obtained by distributing one analysis on $p$ threads, $T_1$ is the execution time on a single thread (sequential run), and $T_p$ is the execution time of the task on $p$ threads [13]. We also compared the parallel performance

**Table 1. Computational time (s) required to complete STRUCTURE and NEWHYBRIDS analyses in series compared to in parallel using EasyParallel, Structure_threader, and Parallelnewhybrid.** The speed gain (in parentheses) was calculated by dividing the execution time on a single thread (sequential run) by the execution time obtained from different number of threads. The analyses were repeated using different operating system and CPU architectures: i7 4700MQ – Lenovo Y510, Windows 10, 2.4 GHz Intel Core i7- 4700MQ with 8 GB RAM and 4 physical cores (8 logical threads); i7 6700HQ – Lenovo Y700, Windows 10, 2.6 GHz Intel Core i7-6700HQ with 8 GB RAM and 4 physical cores (8 logical threads); MacPro i5 – MacBook Pro, OS 10.14, 2.7 GHz Intel Core i5 with 16 GB RAM and 2 physical cores; MacPro i7 – MacBook Pro, OS 10.14, 2.6 GHz Intel Core i7 with 16 GB RAM and 6 physical cores.

| Threads | i7 6700HQ | i7 4700MQ | MacPro i5 | MacPro i7 |
|---|---|---|---|---|
| EasyParallel (STRUCTURE) | | | | |
| 1 | 14711 | 14943 | 8226 | 5307 |
| 2 | 7772 (1.89) | 7929 (1.88) | 4143 (1.99) | 2785 (1.91) |
| 4 | 4052 (3.63) | 5212 (2.87) | – | 1561 (3.40) |
| 6 | 3617 (4.07) | 5106 (2.93) | – | 1300 (4.08) |
| 8 | 3049 (4.82) | 4733 (3.16) | – | – |
| Structure_threader | | | | |
| 1 | 14688 | 14980 | 8193 | 5328 |
| 2 | 7762 (1.89) | 7808 (1.92) | 4145 (1.98) | 2811 (1.90) |
| 4 | 4040 (3.64) | 5255 (2.85) | – | 1551 (3.44) |
| 6 | 3597 (4.08) | 5099 (2.94) | – | 1282 (4.16) |
| 8 | 2999 (4.90) | 4708 (3.18) | – | – |
| EasyParallel (NEWHYBRIDS) | | | | |
| 1 | 1574 | 1594 | 793 | 683 |
| 2 | 810 (1.94) | 820 (1.94) | 418 (1.90) | 375 (1.82) |
| 4 | 489 (3.22) | 606 (2.63) | – | 206 (3.32) |
| 6 | 480 (3.28) | 551 (2.89) | – | 205 (3.33) |
| 8 | 330 (4.77) | 407 (3.92) | – | – |
| Parallelnewhybrid | | | | |
| 1 | 1500 | 1617 | 828 | 710 |
| 2 | 800 (1.87) | 837 (1.91) | 445 (1.86) | 377 (1.88) |
| 4 | 477 (3.08) | 562 (2.81) | – | 208 (3.39) |
| 6 | 478 (3.19) | 553 (2.86) | – | 206 (3.36) |
| 8 | 323 (4.69) | 403 (3.92) | – | – |

between EasyParallel and two existing software, Structure_threader and Parallelnewhybrid, by using the same parameter settings and datasets for parallel analyses. Structure_threader was previously proven to be more efficient and faster than similar multiple-thread methods for performing multiple STRUCTURE runs (StrAuto and ParallelStructure), and therefore was considered an optimal target for performance comparison [13–15]. Parallelnewhybrid was the only known R package designed to execute multiple NEWHYBRIDS runs in parallel [16].

## 3. Results and discussion

For all STRUCTURE and NEWHYBRIDS analyses, the parallel computational time in EasyParallel was faster than a sequential run using a single thread in general (Fig 2, Table 1). However, we note that the speed gain of parallelization was not linear with the increased number of threads. This phenomenon has been previously reported in other parallel programs [13,15,16]. One potential explanation for this nonlinearity is that the operating system and processor must deal with computation resources utilized by intensive tasks (i.e. STRUCTURE and NEWHYBRIDS parallel runs) and underlying system processes, therefore affecting the performance of parallelization [16]. On the other hand, the occurrence of "Cache trashing" may impact the speed of parallelization when working with larger data sets and when both logical threads (in

one physical core) share L2 and L3 caches [15]. However, despite the nonlinearity issue, we observed that the performance of EasyParallel was not limited by the availability of random access memory (RAM), as the usage of RAM was always low during parallelization.

The runtime and speed gain obtained by EasyParallel, Structure_threader, and Parallelnewhybrid were very similar (Fig 2, Table 1), regardless of the number of threads, operating systems, or CPU processors used for the analysis. The same implementation of "multiprocessing" and "subprocess" modules in both EasyParallel and Structure_threader would explain the minimal difference in performance for repeated STRUCTURE running. On the other hand, although EasyParallel and Parallelnewhybrid performed equally well in analyzing multiple simulated data sets, EasyParallel was more efficient in processing the input data, as each thread shared the same executable input file. Parallelnewhybrid, however, needs to duplicate the input data for each thread execution and produce temporary files during parallel computing. Beyond that, the key feature of EasyParallel is its graphical user interface, which facilitates data processing and makes it accessible to users who have limited knowledge in any programming language.

## 4. Conclusion

In summary, we have developed a Python-based software that assists users working with iteration processes in STRUCTURE and NEWHYBRIDS analyses. EasyParallel is a user-friendly and free platform that combines a point-and-click interface and multi-core parallelization for multiple independent runs of the dataset, assisting the user in assessing the most biologically likely *K* and estimating hybrid class assignment accuracy. EasyParallel is also a stand-alone software executable for both MacOS and Windows operating systems, with all modules and the source code packaged for the direct run without installation.

## Acknowledgments

The authors wish to thank Wenlu Wang for code debugging. The authors appreciate the help of Katherine Silliman and Matt Lewis in manuscript revision and in-house program tests.

## Author Contributions

**Conceptualization:** Eric Peatman.

**Formal analysis:** Honggang Zhao.

**Investigation:** Honggang Zhao.

**Methodology:** Honggang Zhao.

**Software:** Honggang Zhao.

**Supervision:** Benjamin Beck, Adam Fuller, Eric Peatman.

**Validation:** Honggang Zhao.

**Writing – original draft:** Honggang Zhao.

**Writing – review & editing:** Honggang Zhao, Benjamin Beck, Adam Fuller, Eric Peatman.

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
