## [Decision Letter · Decision Letter 0]

27 Dec 2019

PONE-D-19-32193

EasyParallel: a GUI platform for parallelization of STRUCTURE and NEWHYBRIDS analyses

PLOS ONE

Dear Dr. Zhao,

Thank you for submitting your manuscript to PLOS ONE. After careful consideration, we feel that it has merit but does not fully meet PLOS ONE’s publication criteria as it currently stands. Therefore, we invite you to submit a revised version of the manuscript that addresses the points raised during the review process.

We would appreciate receiving your revised manuscript by Feb 10 2020 11:59PM. To enhance the reproducibility of your results, we recommend that if applicable you deposit your laboratory protocols in protocols.io, where a protocol can be assigned its own identifier (DOI) such that it can be cited independently in the future. For instructions see: http://journals.plos.org/plosone/s/submission-guidelines#loc-laboratory-protocols

We look forward to receiving your revised manuscript.

Kind regards,

Roberto Fritsche-Neto, Ph.D.

Academic Editor

PLOS ONE

Journal Requirements:

Reviewers' comments:

Reviewer's Responses to Questions

**Comments to the Author**

1. Is the manuscript technically sound, and do the data support the conclusions?

Reviewer #1: Partly

Reviewer #2: Yes

2. Has the statistical analysis been performed appropriately and rigorously? 

Reviewer #1: No

Reviewer #2: Yes

3. Have the authors made all data underlying the findings in their manuscript fully available?

Reviewer #1: Yes

Reviewer #2: Yes

4. Is the manuscript presented in an intelligible fashion and written in standard English?

Reviewer #1: Yes

Reviewer #2: Yes

5. Review Comments to the Author

Reviewer #1: This manuscript has the potential to be useful and talks about one GUI platform for parallelization. Its point-and-click, simple and intuitive platform called EasyParallel. I ran the example data that comes with STRUCTURE and NEWHYBRIDS following the “Demo” available at http://webhome.auburn.edu/~hzz0024/web/demo/ and works pretty well in my computer. However, it’s pretty similar with already exist for parallelization. I really recommend the authors include more functions or tools to let the software more attractive, for example: function to help the users to build the input files for STRUCTURE and NEWHYBRIDS, in other words, a similar approach of questions and answers where the user inform the parameters to get the input files for parallelization. Also, the authors could add a window for results interpretation with, for example graphics and tables, because if the main goal is people with little experience in programming with the actual version of EasyParallel those people still needing other software to process the outputs and make graphics. I tried to insert point-by-point my comments to help in the correction process:

Line 55-57: Its redundant.

Line 70: “ … Monte Carlo (MCMC) to resampling …“

Line 71: I had the impression that the authors are using the word “burnin” as the same idea as total “iterations”. Normally, only the first portion of interactions are called by burnin, where the interaction process exercised the priors. This testing will force failures under supervised conditions and then established the interactions. Could you review this sentence to follow the cited paper Porras-Hurtado et al (2013): “STRUCTURE uses a systematic Bayesian clustering approach applying Markov Chain Monte Carlo (MCMC) estimation. The MCMC process begins by randomly assigning individuals to a pre-determined number of groups, then variant frequencies are estimated in each group and individuals re-assigned based on those frequency estimates. This is repeated many times, typically comprising 100,000 iterations, in the burnin process that results in a progressive convergence toward reliable allele frequency estimates in each population and membership probabilities of individuals to a population.

Measurement of the assumed number of populations uses the MCMC estimation and is performed separately from the burnin.”.

Line 91-92: This is not one of your objectives. In my opinion EasyParallel is doing only the parallelization process and the other software (STRUCTURE and NEWHYBRIDS) are “assisting the user in assessing the most biologically likely number of clusters (K) and estimating hybrid class assignment accurately.”.

Line 128: It is not clear if the “Execution Time Analyses” was performed using the GUI version or code line.

Line 133-138: Why the authors chose these specific machines? Why did not you use computers with i3 or 4GB RAM, that regular people with no experience with programing have?

Line 143: “MCMC” also is an iteration process as “burnin”, please be consistent.

Line 162: “always” it is not true for “i7.6700HQ” and “MacPro i5” during the STRUCTURE comparation. Please use terms as “in general” or “majority”.

Line 171-173: I did not find these results. Could you add a table as a supplementary file with these RAM results? I believe the execution time is being influenced by the fact of MacOS’s computers have twice RAM than Windows’s.

Line 173-177: This comparation should be made with similar machines, same processor (i3, i5, i7, or i9), RAM, physical cores, etc. Also, the comparation of operating systems is not one of your objectives.

Line 195-197: Only in this last sentence is clear that the software doesn’t has the option to run in command code or prompt. Please be more specific and move this part for material and methods.

Figure 1: I confess that I spent a time trying to understand why the EasyParallel logo is a fish. I understand the group works with aquaculture but using a fish as logo is not helping at all to get attention for the software. I really recommend change the logo for some genetic or parallel symbol. In addition, it will be interesting and helpful if the software provides some results visualization as graphics and tables as suggested above.

Figure 2: The authors are comparing computers and not software, which it is the main idea. I recommend to exchange the position of software and computer where the lines should be the software (EasyParallel, STRUCTURE and NEWHYBRIDS).

Table 1: Use “-” instead NA.

The manual is clear and well done. Only one correction in “Step 3” where the mainparams was wrote twice at the link: http://webhome.auburn.edu/~hzz0024/web/doc/.

Reviewer #2: This paper presented a Python-based software named EasyParallel that assists users working with iteration processes in STRUCTURE and NEWHYBRIDS analyses. STRUCTURE and NEWHYBRIDS software are widely used in population genetic structure studies, admixture, and hybridization. The analyzes performed by these programs usually require a large computational time, especially when large genotyped populations with a large number of molecular markers are analyzed. The multi-threaded parallelization of EasyParallel allows processing of large genetic datasets in an efficient way, providing ready access to users who have little experience in script programming. The authors use clear and straightforward language and provide all relevant data and information. Therefore, I recommend accepting the article for publication.

6. PLOS authors have the option to publish the peer review history of their article (what does this mean?). If published, this will include your full peer review and any attached files.

Reviewer #1: No

Reviewer #2: No

---

## [Author Response · Author response to Decision Letter 0]

10 Mar 2020

PONE-D-19-32193

EasyParallel: a GUI platform for parallelization of STRUCTURE and NEWHYBRIDS analyses

PLOS ONE

Dear Dr. Zhao,

Thank you for submitting your manuscript to PLOS ONE. After careful consideration, we feel that it has merit but does not fully meet PLOS ONE’s publication criteria as it currently stands. Therefore, we invite you to submit a revised version of the manuscript that addresses the points raised during the review process.

We would appreciate receiving your revised manuscript by Feb 10 2020 11:59PM. To enhance the reproducibility of your results, we recommend that if applicable you deposit your laboratory protocols in protocols.io, where a protocol can be assigned its own identifier (DOI) such that it can be cited independently in the future. For instructions see: http://journals.plos.org/plosone/s/submission-guidelines#loc-laboratory-protocols

• A rebuttal letter that responds to each point raised by the academic editor and reviewer(s). This letter should be uploaded as separate file and labeled 'Response to Reviewers'.

• A marked-up copy of your manuscript that highlights changes made to the original version. This file should be uploaded as separate file and labeled 'Revised Manuscript with Track Changes'.

• An unmarked version of your revised paper without tracked changes. This file should be uploaded as separate file and labeled 'Manuscript'.

We look forward to receiving your revised manuscript.

Kind regards,

Roberto Fritsche-Neto, Ph.D.

Academic Editor

PLOS ONE

Journal Requirements:

Reviewers' comments:

Reviewer's Responses to Questions

Comments to the Author

1. Is the manuscript technically sound, and do the data support the conclusions?

Reviewer #1: Partly

Reviewer #2: Yes

2. Has the statistical analysis been performed appropriately and rigorously?

Reviewer #1: No

Reviewer #2: Yes 

3. Have the authors made all data underlying the findings in their manuscript fully available?

Reviewer #1: Yes

Reviewer #2: Yes

4. Is the manuscript presented in an intelligible fashion and written in standard English?

Reviewer #1: Yes

Reviewer #2: Yes 

5. Review Comments to the Author

Reviewer #1: This manuscript has the potential to be useful and talks about one GUI platform for parallelization. Its point-and-click, simple and intuitive platform called EasyParallel. I ran the example data that comes with STRUCTURE and NEWHYBRIDS following the “Demo” available at http://webhome.auburn.edu/~hzz0024/web/demo/ and works pretty well in my computer. However, it’s pretty similar with already exist for parallelization. 

We appreciate reviewer’s comments here. We agree that programs such as Structure_threader and parallelnewhybrid have been developed to help process tasks in parallel on a multi-threaded processor. However, such software requires minimum programming knowledge (e.g., R, Bash) for program running. To our best knowledge, our program is the first GUI platform that supports parallel running of STRUCTURE and NEWHYBRIDS.

I really recommend the authors include more functions or tools to let the software more attractive, for example: function to help the users to build the input files for STRUCTURE and NEWHYBRIDS, in other words, a similar approach of questions and answers where the user inform the parameters to get the input files for parallelization. Also, the authors could add a window for results interpretation with, for example graphics and tables, because if the main goal is people with little experience in programming with the actual version of EasyParallel those people still needing other software to process the outputs and make graphics. I tried to insert point-by-point my comments to help in the correction process:

While we appreciate the reviewer’s suggestion here, we feel that it requires a careful design and lots of efforts to build a function for input manipulations, as the STRUCTURE itself needs two parameter files (mainparams and extraparams) along with the genotype input data. Both STRUCTURE (https://web.stanford.edu/group/pritchardlab/structure_software/release_versions/v2.3.4/structure_doc.pdf) and NEWHYBRIDS (https://github.com/eriqande/newhybrids/blob/master/new_hybs_doc1_1Beta3.pdf) did excellent jobs in documenting the input parameters. Besides, some existing programs such as widgetcon (Aydın et al., 2019), and PGDSpider (Lischer and Excoffier, 2012) have been well developed to prepare the input data. Therefore, we feel our way of presenting the front end is the most clear for readers in parallel computing, and in the future we will design such functions as suggested by the reviewer. 

We agree that it would be helpful to develop a window for results interpretation and plotting. However, we found that some existing programs such as CLUMPAK (Kopelman et al., 2015), POPHELPER (Francis, 2017), StructureSelector (Li and Liu, 2017), and KFinder (Wang, 2019) have been widely adopted for output plotting and interpretation. We will consider reviewer’s advice and add this function in our next release.

Line 55-57: Its redundant.

We have rephased this sentence to make it simple and clean.

Line 70: “ … Monte Carlo (MCMC) to resampling …“

Corrected

Line 71: I had the impression that the authors are using the word “burnin” as the same idea as total “iterations”. Normally, only the first portion of interactions are called by burnin, where the interaction process exercised the priors. This testing will force failures under supervised conditions and then established the interactions. Could you review this sentence to follow the cited paper Porras-Hurtado et al (2013): “STRUCTURE uses a systematic Bayesian clustering approach applying Markov Chain Monte Carlo (MCMC) estimation. The MCMC process begins by randomly assigning individuals to a pre-determined number of groups, then variant frequencies are estimated in each group and individuals re-assigned based on those frequency estimates. This is repeated many times, typically comprising 100,000 iterations, in the burnin process that results in a progressive convergence toward reliable allele frequency estimates in each population and membership probabilities of individuals to a population.Measurement of the assumed number of populations uses the MCMC estimation and is performed separately from the burnin.”.

We have rephased the sentence as “Because STRUCTURE requires to minimize the effect of the starting configuration, many iterations are needed during the burnin process.

Line 91-92: This is not one of your objectives. In my opinion EasyParallel is doing only the parallelization process and the other software (STRUCTURE and NEWHYBRIDS) are “assisting the user in assessing the most biologically likely number of clusters (K) and estimating hybrid class assignment accurately.”

We agree with it. Our manuscript explicitly explains our platform builds upon STRUCTURE and NEWHYBRIDS. What we tried to explain here is that our platform enables running multiple Ks (K within the predefined range) using one-click, thus facilitating the user in assessing the optimal K without running the program multiple times with different K repeatedly. We have addressed reviewer’s comment by deleting this sentence.

Line 128: It is not clear if the “Execution Time Analyses” was performed using the GUI version or code line.

The analyses are based on the GUI version, and we revised the manuscript to clarify the setting from line 97-99.

Line 133-138: Why the authors chose these specific machines? Why did not you use computers with i3 or 4GB RAM, that regular people with no experience with programing have?

Thank you for your valuable advice. We were trying to test on as many machines as possible, and test our performance in various settings. However, we decided to focus on forward compatibility instead of backward compatibility. We will try to cover more different settings in our next release.

Line 143: “MCMC” also is an iteration process as “burnin”, please be consistent.

Corrected

Line 162: “always” it is not true for “i7.6700HQ” and “MacPro i5” during the STRUCTURE comparation. Please use terms as “in general” or “majority”.

We have revised the manuscript accordingly.

Line 171-173: I did not find these results. Could you add a table as a supplementary file with these RAM results? I believe the execution time is being influenced by the fact of MacOS’s computers have twice RAM than Windows’s.

We appreciate reviewer’s valuable comment here. However, we made this conclusion only by observing the real-time RAM usage and did not record such data. STRUCTURE and NEWHYBRIDS are not memory demanding algorithms, and the size of RAM is not a bottleneck for parallelization. Same observation has been also reported in Besnier et al. (2013) and Wringe et al. (2017). 

Line 173-177: This comparation should be made with similar machines, same processor (i3, i5, i7, or i9), RAM, physical cores, etc. Also, the comparation of operating systems is not one of your objectives.

We agree with reviewer that such comparisons should be made using same machinery settings. We have revised the manuscript accordingly. 

Line 195-197: Only in this last sentence is clear that the software doesn’t has the option to run in command code or prompt. Please be more specific and move this part for material and methods.

We appreciate reviewer’s comment here. We have state this in the material and methods part (line 97-99). We hope it clarifies our intention. 

Figure 1: I confess that I spent a time trying to understand why the EasyParallel logo is a fish. I understand the group works with aquaculture but using a fish as logo is not helping at all to get attention for the software. I really recommend change the logo for some genetic or parallel symbol. In addition, it will be interesting and helpful if the software provides some results visualization as graphics and tables as suggested above.

We appreciate reviewer’s valuable comment and we do hope the software could be used in a broad community. We followed reviewer’s comment and redesigned the EasyParallel logo to make it easy to remember and reflect the parallelization. Again, we appreciate reviewer’s advice about results visualization. Please see our answer at the start of our response.

Figure 2: The authors are comparing computers and not software, which it is the main idea. I recommend to exchange the position of software and computer where the lines should be the software (EasyParallel, STRUCTURE and NEWHYBRIDS).

We agree with reviewer and redrew the Figure 1

.

Table 1: Use “-” instead NA.

Corrected

The manual is clear and well done. Only one correction in “Step 3” where the mainparams was wrote twice at the link: http://webhome.auburn.edu/~hzz0024/web/doc/.

Corrected

Reviewer #2: This paper presented a Python-based software named EasyParallel that assists users working with iteration processes in STRUCTURE and NEWHYBRIDS analyses. STRUCTURE and NEWHYBRIDS software are widely used in population genetic structure studies, admixture, and hybridization. The analyzes performed by these programs usually require a large computational time, especially when large genotyped populations with a large number of molecular markers are analyzed. The multi-threaded parallelization of EasyParallel allows processing of large genetic datasets in an efficient way, providing ready access to users who have little experience in script programming. The authors use clear and straightforward language and provide all relevant data and information. Therefore, I recommend accepting the article for publication. 

We appreciate reviewer’s comments and recommendation!

6. PLOS authors have the option to publish the peer review history of their article (what does this mean?). If published, this will include your full peer review and any attached files.

Do you want your identity to be public for this peer review? For information about this choice, including consent withdrawal, please see our Privacy Policy.

Reviewer #1: No

Reviewer #2: No

References:

Aydın, M., Kryvoruchko, I. S., & Şakiroğlu, M. (2019). widgetcon: A website and program for quick conversion among common population genetic data formats. Molecular Ecology Resources, 19(5), 1374-1377.

Besnier, F., & Glover, K. A. (2013). ParallelStructure: AR package to distribute parallel runs of the population genetics program STRUCTURE on multi-core computers. PLoS One, 8(7).

Francis, R. M. (2017). pophelper: an R package and web app to analyse and visualize population structure. Molecular Ecology Resources, 17(1), 27-32.

Li, Y. L., & Liu, J. X. (2018). StructureSelector: A web‐based software to select and visualize the optimal number of clusters using multiple methods. Molecular Ecology Resources, 18(1), 176-177.

Lischer, H. E., & Excoffier, L. (2012). PGDSpider: an automated data conversion tool for connecting population genetics and genomics programs. Bioinformatics, 28(2), 298-299.

Kopelman, N. M., Mayzel, J., Jakobsson, M., Rosenberg, N. A., & Mayrose, I. (2015). Clumpak: a program for identifying clustering modes and packaging population structure inferences across K. Molecular Ecology Resources, 15(5), 1179-1191.

Wang, J. (2019). A parsimony estimator of the number of populations from a STRUCTURE‐like analysis. Molecular Ecology Resources, 19(4), 970-981.

Wringe, B. F., Stanley, R. R., Jeffery, N. W., Anderson, E. C., & Bradbury, I. R. (2017). parallelnewhybrid: an R package for the parallelization of hybrid detection using newhybrids. Molecular Ecology Resources, 17(1), 91-95.

---

## [Decision Letter · Decision Letter 1]

8 Apr 2020

EasyParallel: a GUI platform for parallelization of STRUCTURE and NEWHYBRIDS analyses

PONE-D-19-32193R1

Dear Dr. Zhao,

We are pleased to inform you that your manuscript has been judged scientifically suitable for publication and will be formally accepted for publication once it complies with all outstanding technical requirements.

With kind regards,

Roberto Fritsche-Neto, Ph.D.

Academic Editor

PLOS ONE

Additional Editor Comments (optional):

Reviewers' comments:

Reviewer's Responses to Questions

**Comments to the Author**

1. If the authors have adequately addressed your comments raised in a previous round of review and you feel that this manuscript is now acceptable for publication, you may indicate that here to bypass the “Comments to the Author” section, enter your conflict of interest statement in the “Confidential to Editor” section, and submit your "Accept" recommendation.

Reviewer #1: All comments have been addressed

2. Is the manuscript technically sound, and do the data support the conclusions?

Reviewer #1: Yes

3. Has the statistical analysis been performed appropriately and rigorously? 

Reviewer #1: Yes

4. Have the authors made all data underlying the findings in their manuscript fully available?

Reviewer #1: Yes

5. Is the manuscript presented in an intelligible fashion and written in standard English?

Reviewer #1: Yes

6. Review Comments to the Author

Reviewer #1: Thank you to accept my suggestions.

The only minor revision it is to change the logo at https://github.com/hzz0024/EasyParallel.

7. PLOS authors have the option to publish the peer review history of their article (what does this mean?). If published, this will include your full peer review and any attached files.

Reviewer #1: Yes: Filipe Inácio Matias

---

## [Editor Report · Acceptance letter]

13 Apr 2020

PONE-D-19-32193R1 

EasyParallel: a GUI platform for parallelization of STRUCTURE and NEWHYBRIDS analyses 

Dear Dr. Zhao:

I am pleased to inform you that your manuscript has been deemed suitable for publication in PLOS ONE. Congratulations! Your manuscript is now with our production department. 

With kind regards,

on behalf of

Professor Roberto Fritsche-Neto 

Academic Editor

PLOS ONE